# Evaluation of Fetal Cardiac Geometry and Contractility in Gestational Diabetes Mellitus by Two-Dimensional Speckle-Tracking Technology

**DOI:** 10.3390/diagnostics12092053

**Published:** 2022-08-24

**Authors:** Roxana Gireadă, Demetra Socolov, Elena Mihălceanu, Ioan Tudor Lazăr, Alexandru Luca, Roxana Matasariu, Alexandra Ursache, Iuliana Bujor, Tiberiu Gireadă, Vasile Lucian Boiculese, Răzvan Socolov

**Affiliations:** 1Department of Obstetrics and Gynecology, University of Medicine and Pharmacy ‘Gr. T. Popa’, 700115 Iaşi, Romania; 2Department of Obstetrics and Gynecology, Cuza Vodă Hospital, 700038 Iaşi, Romania; 3Department of Medical Informatics and Biostatistics, University of Medicine and Pharmacy ‘Gr. T. Popa’, 700115 Iaşi, Romania; 4Department of Obstetrics and Gynecology, Elena Doamna Hospital, 700398 Iaşi, Romania

**Keywords:** gestational diabetes mellitus, fetal cardiac function, echocardiography, two-dimensional speckle-tracking, FetalHQ^®^

## Abstract

*Background:* The most commonly known cardiac effect of gestational diabetes mellitus (GD) in the fetus is hypertrophic cardiomyopathy, but recent studies show that it is preceded by subclinical cardiac dysfunction. This study aimed to assess the effect of GD on fetal cardiac geometry and contractility by two-dimensional speckle-tracking technology. *Methods:* We performed a prospective observational study that included 33 pregnant patients with GD and 30 healthy individuals. For all fetuses, a four-chamber 3 s cine-loop was recorded and analyzed with Fetal Heart Quantification (FetalHQ^®^), a novel proprietary speckle-tracking software. The following cardiac indices were calculated: global sphericity index (GSI), global longitudinal strain (GLS), fractional area change (FAC), and 24-segment end-diastolic diameter (EDD), fractional shortening (FS), and sphericity index (SI) for both ventricles. Demographic and cardiac differences between the two groups were analyzed, as well as intra-rater and inter-rater reliability. *Results:* There were significant changes in right ventricular FAC and FS for segments 4–24 in fetuses exposed to GD (−1 SD, *p* < 0.05). No significant differences were detected for GSI, GLS, EDD, or SI for either ventricle. *Conclusions:* Fetuses exposed to GD present impaired right ventricular contractility, especially in the mid and apical segments.

## 1. Introduction

Gestational diabetes mellitus (GD) represents an impaired glucose metabolism, whose definition shifted in time from any degree of glucose intolerance first recognized during pregnancy to hyperglycemia recognized in the second or third trimester of pregnancy, in a patient without clearly overt diabetes prior to gestation and with a normal provoked hyperglycemia test in the postnatal period [1]. According to the latter definition, the weighted prevalence of GD in Europe is 17.1% [2]. GD prevalence is also significantly higher in women over 30 years and in overweight/obese women [2]. Thus, GD is a serious health problem, especially considering the increasing maternal age at childbirth and the global obesity epidemic.

GD puts the mother, fetus, and newborn at increased risk of adverse outcome, dependent on the maternal glycemia at 24–28 weeks of gestation, without threshold for most complications [3]. The risk of intrauterine fetal demise is increased five-fold in GD, and the cause cannot be determined in 50% of cases [4]. Several explanations have been proposed: macrosomia due to fetal hyperinsulinemia, excess fat deposition, increased metabolism with fetal hypoxemia [4], and even cardiac dysfunction [5]. However, to present, there are no efficient tools to predict and prevent GD-related stillbirth, since it can occur even in well-controlled diabetes [5]. In this regard, subclinical cardiac dysfunction could be a lucrative research avenue, since recent studies have shown that fetuses exposed to GD exhibit impaired contractility before cardiac hypertrophy [6,7,8] and despite good glycemic control [9]. The pathophysiology of these cardiac changes is not fully clear, but several mechanisms have been incriminated (Figure 1). Although insulin does not cross the placenta, GD-associated maternal hyperinsulinemia could affect the fetus through placental insulin receptor mediation, while glucose passing freely through the placenta into the fetal circulation could stimulate fetal pancreatic ß-cells [10]. Thus, GD associates simultaneous maternal and fetal hyperinsulinemia, with potential deleterious effects on the fetal cardiac function [10]. Additionally, animal studies have found that hyperglycemia is linked to an excess of reactive oxygen species [11]. The degree of oxidative stress correlates with maternal glycemic control and interventricular septum (IVS) hypertrophy in diabetes-exposed neonates [12]. Bradley et al. found significant acidosis and increased lactate levels in the umbilical cord of diabetes-exposed fetuses and suggested this hypoxemic environment could explain stillbirth [13]. Moreover, Lehtoranta et al. proved that expression of key genes involved in cardiomyocyte electrophysiology, contractility, and metabolism are altered in rat models of fetuses exposed to maternal hyperglycemia, and suggested this leads to cardiac dysfunction and increased fetal vulnerability to hypoxemia [14].

While much remains to be known about the underlying mechanism, cardiac dysfunction can be evaluated in GD fetuses through novel techniques adapted from adult cardiology. Speckle-tracking technology is a promising tool for assessing fetal cardiac geometry and contractility, since it has several advantages: it is less angle-dependent, proprietary software has recently become available directly on the ultrasound machines [15], and it is better at detecting mild cardiac dysfunction and at predicting adverse outcomes compared to conventional techniques [7,16]. So far, speckle tracking has been used to evaluate diabetes-exposed fetuses for research purposes only. Most studies have proven fetal cardiac function is impaired, but they do not report the exact same changes [7,8,9,17,18,19,20,21,22].

The aim of this study was to assess the effect of GD on fetal cardiac geometry and contractility by using Fetal Heart Quantification (FetalHQ^®^), a two-dimensional speckle-tracking technology available on General Electric Voluson^®^ ultrasound machines. Our primary objective was to evaluate the presence of cardiac changes in GD fetuses, and the secondary objective was to evaluate the reliability of FetalHQ^®^ measurements.

## 2. Materials and Methods

### 2.1. Study Population

This prospective observational study was conducted on pregnant patients diagnosed with GD, in comparison to healthy individuals. Patients were recruited from pregnant women attending prenatal routine care during their third trimester in a private setting from Iaşi, Romania, between June 2021–May 2022 (Figure 1).

The inclusion criteria were as follows: maternal age ≥ 18 years, singleton fetus with normal growth and anatomy, first-trimester dating by crown-rump-length measurement at 11–13^+6^ weeks of gestation, fasting 75 g 2-h oral glucose tolerance test (OGTT) performed between 24–28 weeks of gestation, and known pregnancy outcome. The exclusion criteria were as follows: fetal structural or chromosomal abnormalities, pre-existing diabetes, maternal or obstetrical pathology other than GD, adverse pregnancy outcome unrelated to GD, and poor quality imaging.

GD was defined as diabetes diagnosed in the second or third trimester of pregnancy that was not clearly overt diabetes prior to gestation [1]. The diagnosis of GD was made by a positive 75 g OGTT between 24–28 weeks of gestation, that is, at least one of the venous glucose values was above the normal threshold: fasting ≥ 92 mg/dL, 1 h postprandial ≥ 180 mg/dL, 2 h postprandial ≥ 153 mg/dL [1]. Patients with a positive OGTT were included in the GD group; individuals with a negative OGTT at 24–28 weeks of gestation and no suspicion of GD throughout the remainder of the pregnancy were included in the control group. All GD patients were referred to a diabetologist for counseling and follow-up. GD was considered poorly controlled if ≥30% of capillary plasma glucose values were abnormal in the 10 days preceding the scan (fasting ≥ 95 mg/dL, 1 h postprandial ≥ 140 mg/dL, 2 h postprandial ≥ 120 mg/dL) [23]. If glycemic control was not obtained after 10 days of diet and exercise, the patient was offered insulin treatment. Macrosomia was defined by an abdominal circumference ≥ 97.5% and/or estimated fetal weight ≥ 97.5%. Hydramnios was defined by the deepest vertical pocket of amniotic fluid being greater than 10 cm.

Data about demographics, ultrasound findings, and pregnancy outcome were collected during the prenatal consultations and by contacting the patients by phone after delivery.

### 2.2. Echocardiography

The scans were performed transabdominally by one specifically trained sonographer (R.G.), using a Voluson^®^ E10 BT19 Ultrasound System (General Electric Healthcare, Milwaukee, WI, USA), wideband convex volume probe RM7C, 2–8 MHz.

All scans for the GD group were paired with scans from healthy individuals, performed at the same gestational age (GA) ± 2 days.

The first step was to assess fetal wellbeing, including the amniotic fluid evaluation and estimation of the fetal weight by Hadlock’s formula [24]. For speckle-tracking analysis, a 3 s two-dimensional cine-loop of the 4-chamber (4C) view was recorded using the 2/3 Trimester Cardiac module, with the IVS oriented horizontally or obliquely. The recording was made in the absence of maternal or fetal movements, adjusting the width, depth, focus, and tissue harmonic function to obtain the highest frame rate possible. The gain was adjusted to achieve optimal endocardial border visualization. The standard allotted time for scanning was 30 min, with pausing and rescanning if necessary; if an adequate cine-loop was not obtained within a 90-min interval, the patient was excluded from the study.

After anonymization, all cine-loops were analyzed using FetalHQ^®^, a proprietary two-dimensional speckle-tracking software (General Electric Medical Systems, Milwaukee, WI, USA). For the GD group, the last ultrasound examination prior to delivery was used for the analysis and paired with a GA-matched examination from the control group. The highest quality cine-loop was chosen, and then FetalHQ^®^ measurements were selected.

The global size and shape of the heart were assessed by measuring the 4C end-diastolic (ED) length and width, with the software computing the global sphericity index (GSI) by dividing 4C ED length by 4C ED width [25].

To evaluate the ventricular shape and contractility, a cardiac cycle was selected by using the M-mode—a line was drawn perpendicularly to the IVS (when IVS was perpendicular to the ultrasound beam) or to the tricuspid valve (when the IVS was oblique to the ultrasound beam) (Appendix A). The software automatically generated an M-mode image, and the clearest cardiac cycle was chosen by setting the first ED, the consecutive ED, and finally, the end-systole (ES). The cine-loop was visually assessed while rolling the trackball, to select the cardiac cycle with the best endocardial visualization and for a better definition of the ED and ES time points (ED corresponded to the largest ventricular area and ES corresponded to the smallest ventricular area). After choosing the cardiac cycle, the ES endocardial border of the left ventricle (LV) was defined by 3 points indicated by the operator (1—the intersection between the atrioventricular valve and the septal wall, 2—the intersection between the atrioventricular valve and the lateral wall, 3—the apex). The system automatically generated extra tracing points for the endocardial border. A 4C cine-loop with superimposed tracking curves could be zoomed to check that the endocardium was adequately traced, and fine-tuning was performed by the operator (the ED tracing could be adjusted independently of the ES tracing but changing the latter also impacted former). This process was then repeated for the right ventricle (RV). After the observer made all the adjustments, the ‘Add to Report & Exit’ button was selected, and data were exported as a FetalHQ^®^ (CSV) file. Although aware of the aim of the study, the operator could not repeat the measurements after seeing the Z-scores graphs.

After tracking the endocardial border, the software computed the following cardiac parameters: the global longitudinal strain (GLS) for both ventricles (GLS = (ES endocardial length − ED endocardial length) × 100/ ED endocardial length) [26]; fractional area change (FAC) for both ventricles (FAC = (ED area  − ES area) × 100/ED area) [27]; LV ejection fraction (EF = (ED volume − ES volume) × 100/ED volume); LV stroke volume (SV = ED volume − ES volume); LV cardiac output (CO = SV × heart rate); 24-segment end-diastolic diameter (EDD); 24-segment transverse fractional shortening (FS = (EDD − ES diameter) × 100/EDD) [28]; and 24-segment sphericity index (SI = ED length/ED transverse diameter) [29] for both ventricles, in absolute values and Z-score. The selected variables for Z-score calculation were the clinical GA (as determined by the first-trimester crown-rump-length), abdominal circumference (Hadlock %) and estimated fetal weight (Hadlock %).

### 2.3. Intra- and Inter-Rater Reliability

The same cardiac cycle was remeasured by the first observer (R.G.) after a 2 to 6-week interval, and by a second observer (T.G.) on 16 randomly chosen echocardiograms. Cardiac cycles were ranked on the M-mode display, and the operator was instructed to select the same cardiac cycle by indicating its rank, so that remeasuring was blinded. Intra-class correlation coefficient (ICC) was assessed to determine intra- and inter-rater reliability.

### 2.4. Statistical Analysis

We performed a descriptive statistical analysis using the Statistical Package for the Social Sciences (version 28.0.1.1, IBM Corp., Armonk, NY, USA). Categorical variables were counted and expressed as frequencies. Continuous variables were expressed as mean ± standard deviation (SD). To check for significant differences between the two groups (GD versus control group), we used the chi-square or Fisher test for categorical variables. Continuous variables were tested for normality using the Shapiro–Wilk test. If they were normally distributed (*p* > 0.05), the independent-samples *t*-test was applied to check for significant differences; the Mann–Whitney test was used for non-normally distributed data. A two-tailed *p*-value < 0.05 was considered significant.

ICC estimates and their 95% confidence intervals were computed using Excel (Microsoft Office 2019 Professional Plus, Microsoft Corporation, Redmond, WA, USA) and the Real Statistics Resource Pack (Release 8.2.1, Charles Zaiontz, Milan, Italy) [30]. For intra-rater ICC we used a mean-rating, absolute agreement, two-way mixed model; for inter-rater ICC we used a mean-rating, absolute agreement, two-way random model. ICC values between 0.5–0.75 defined moderate reliability, ≥0.75 defined good reliability, and ≥0.9 defined excellent reliability.

### 2.5. Ethical Approval

The study was approved by the Ethics Committee of the University of Medicine and Pharmacy ‘Gr. T. Popa’, Iaşi (06/2 September 2020), and all participants gave their written informed consent.

## 3. Results

The final case group included 33 GD patients, matched with 33 ultrasound examinations from 30 healthy individuals (3 control patients were scanned twice so that each GD examination was provided with an adequate GA match).

### 3.1. Study Population Characteristics

The demographic and ultrasound technical characteristics are presented in Table 1. Maternal characteristics were mostly similar between the GD and control groups, except for the body mass index at conception, which was slightly higher for the GD group (24.5 versus 22.3 kg/m^2^).

GD was poorly controlled in thirteen of thirty-three cases (39.4%), even after starting insulin treatment in two cases. Diet adherence was generally not strict, and eleven of thirteen patients refused insulin treatment. There was no fetal intrauterine demise in the two groups. Macrosomia and hydramnios were more frequent in the GD fetuses, as well as an earlier delivery GA, but of no clinical relevance (38.6 versus 39.1 weeks).

Adequate imaging was not possible in only one GD case, due to the fetal lie (cardiac apex persisted at 6 a.m.). As for the ultrasound technical conditions, the depth was slightly higher in the GD group (10.4 versus 9.8 cm), but the pixel size was the same (0.106 mm) and the difference of fetal heart rate to frame rate ratio was clinically irrelevant (2 versus 2.2 bpm/Hz). We achieved a frame rate above 80 Hz in 12% of the GD scans and 9% of the control scans.

### 3.2. Fetal Cardiac Geometry and Contractility

There were no significant differences in global cardiac geometry (similar GSI for both groups). As for the biventricular function analysis, there were no significant differences for LV GLS, FAC, EF, SV, or CO. The RV GLS was not significantly different between the GD and healthy fetuses. The FAC was significantly lower for the RV (33.27 versus 40.33%, *p* < 0.001; Z-GA score −1.32 versus 0.01; *p* < 0.001), and RV ES area was higher (1.8 versus 1.56 cm^2^, *p* = 0.035) (Appendix A).

As for the 24-segment analysis, for the LV there was no significant difference between the 2 groups for the EDD (Appendix A), SI (Appendix A), or FS (Appendix A). The transverse FS was significantly lower for segments 4–24 of the RV (Table 2, Figure 2), but there were no significant differences detected for the RV EDD (Appendix A) or SI (Appendix A).

### 3.3. Intra-Rater and Inter-Rater Reliability

Table 3 presents the intra and inter-rater reliability. For absolute value measurements like ED and ES area, and ED and ES length, the intra-rater reliability was good to excellent (>0.75), while the inter-rater reliability was moderate to good (0.5–0.9). GLS and FAC showed good reliability for the LV, but the ICC was much lower for the RV. For the 24-segment EDD and SI analysis, the intra-rater and inter-rater reliability were mostly good. For the 24-segment FS, the intra-rater and inter-rater reliability were mostly moderate. Compared to the RV, the intra-rater and inter-rater ICC were higher for most LV measurements.

## 4. Discussion

There is increasing evidence that in-utero exposure to GD affects the fetal heart, not only in the form of hypertrophic cardiomyopathy [31,32], but also as subclinical cardiac dysfunction, in rat models [14] as well as in humans [32]. According to a meta-analysis published by Depla et al., diastolic and global cardiac function are decreased in GD, but the effect on systolic function is inconclusive [32]. Recent technological advances have opened the door for more in-depth analysis of fetal cardiac deformation. Our study evaluated the systolic function in GD fetuses compared to healthy fetuses, with the aid of two-dimensional speckle-tracking echocardiography. The main finding of this study is that GD-exposed fetuses have decreased RV transverse contractility, especially in the mid and apical segments.

### 4.1. Cardiac Geometry and Contractility in GD-Exposed Fetuses

The cardiac shape is a good indicator of cardiac function [25]. Some studies reported rounder hearts in GD fetuses [20,21,22], but novel studies using FetalHQ^®^ to compare GD to healthy fetuses did not find a significant difference in GSI [8,9], which is consistent with our findings.

Strain is another sensitive marker of cardiac function. Longitudinal strain is used more often than radial or circumferential strain to evaluate myocardial deformation [21]. There are conflicting results regarding fetal cardiac strain in maternal diabetes, which could be explained by the different study design (most studies did not differentiate pregestational diabetes of GD effects, while some studies used the older GD diagnosis criteria). However, most studies reported decreased RV and/or LV GLS in fetuses exposed to gestational and/or pregestational diabetes [7,17,18,20], and it has also been shown that decreased LV GLS is present after birth in infants of diabetic mothers [33]. Kulkarni et al. proved there is decreased LV strain for all three main directions (longitudinal, circumferential, and radial) even when adjusting for obesity, arguing this impairment indicates a diffuse pattern of myocardial involvement and hypothesizing a myopathic mechanism rather than an adaptive one [17]. Miranda et al. found a decrease of RV (but not LV) global and diastolic longitudinal strain with each additional year of maternal age, predicting a decrease of −0.24% (95% CI, −0.43% to −0.045%) in RV-GLS, independently of maternal body mass index, smoking, and multiparity [7]. Moreover, IVS thickness had no correlation with LV or RV GLS [7]. Yovera et al. studied the effects of diabetes on fetal cardiac function at 24 + 0 to 32 + 0 weeks and at 32 + 1 and 40 + 1 weeks, and found significant changes for both intervals: reduced RV (but not LV) GLS, reduced basal, mid, and apical RV longitudinal strain, and reduced basal LV longitudinal strain [20]. Rolf et al. reported decreased LV and two-chamber GLS and high interventricular dyssynchrony [18]. Contrary to these studies, Patey et al. demonstrated increased LV and RV GLS, therefore increased contractility, but this reversed after birth [19]. The fact that these studies do not report the exact same changes could stem from different GD diagnosis criteria and type of diabetes included in the study, with type 1 diabetes and pregestational type 2 diabetes presumably having more profound effects. Moreover, extensive cardiac deformation analysis has become readily available to sonographers only recently (FetalHQ^®^ is available on Voluson E10 ultrasound machines since the end of 2018) [15]. Studies that applied FetalHQ^®^ only to GD fetuses proved decreased LV and RV GLS [8,9,21], as early as 24 weeks [21], but our study did not replicate these results.

Because of its triangular, crescent shape, RV EF cannot be computed as easily as for the LV, but FAC can serve as a good surrogate for RV systolic function assessment [27]. In our study, FAC was significantly lower for the RV (Z-score difference −1.31 SD), which is consistent with recent studies [21,22].

The novelty of FetalHQ^®^ consists in the 24-segment analysis. To our knowledge, there are only three other studies that reported a 24-segment analysis in GD fetuses [8,9,21], and our findings are fairly consistent with these reports: GD mainly affects the RV in the mid and apical segments, and changes are more prevalent for the transverse than for the global or longitudinal contractility. In our study, there was no significant difference for the EDD, which aligns with previous findings [21]. Twenty-four segment SI could be used throughout gestation since it does not significantly correlate with fetal biometry or GA [29]. In our study, there were no significant changes for LV or RV 24-segment SI, which aligns with the results published by Huang et al. [8] and Chen et al. [9]. Wang et al. found statistically significant lower SIs for LV segments 1–6 and RV segments 5–13 of GD fetuses, but this difference was of small clinical relevance, of no more than 0.5 SD [21]. Our study showed impaired transverse contractility for RV segments 4–24, as reflected by a FS Z-score difference of −1 SD. When interpreting cardiac deformation results, one must pay attention to the following factors: region of interest definition or measurement location (endocardial, epicardial, myocardial midline, full wall), and segment definition (for apical or short-axis view) [26]. The 24-segment FS determined by FetalHQ^®^ is a radial cardiac deformation parameter measured at the endocardium level in a 4C view (equivalent to the apical view in adult cardiology), and it reflects systolic function. Since EDD was seemingly unaffected, decreased RV FS for segments 4–24 clearly demonstrates systolic RV dysfunction in GD-exposed fetuses. This is consistent with recent cardiac deformation studies that used a similar methodology [8,9]. Although a meta-analysis performed in 2021 did not find a conclusive deleterious effect of GD on fetal systolic function [32], it included studies that reported FS measurements relying mostly on the LV [7,17,19,34,35,36,37,38,39,40,41], and the evaluated RV segment was usually basal or unspecified [19,34,35,36,37].

In order to explain RV dysfunction, one must first understand the microarchitecture, biomechanics, and metabolism of the fetal RV. As opposed to the LV three-layered structure, the RV is composed of only two layers: the deep RV layer contains longitudinal fibers, while the superficial layer contains circumferential fibers that orient obliquely towards the RV apex to then continue into the LV superficial layer [42]. The longitudinal shortening ensures 75% of the RV contraction, which is further completed by the shortening of the horizontal fibers, and RV torsion is almost non-existent compared to the LV [42]. The fetal RV functions as a low-pressure systemic ventricle, accounting for 2/3 of the combined cardiac output [43], and the RV free wall has a higher regional blood flow than the LV [42]. Another particularity is that fetal cardiomyocytes rely on glycolysis to produce energy, but their metabolism shifts after birth to fatty acid oxidation as a result of increasing partial oxygen pressure [44]. In conclusion, the RV is the dominant ventricle in fetal life, and it relies on glucose metabolism. Therefore, it is expected that GD would affect the RV first. However, interpreting RV dysfunction in the fetus is complicated by the fact that the two ventricles work in parallel and there are several shunts through which the fetal heart can ‘unload’ during stressful states. Moreover, although longitudinal shortening accounts for most of the RV contraction, Chen et al. showed more frequent changes in the transverse rather than the longitudinal contractility [9], which is also supported by our findings. Contractility depends on preload, afterload, heart rate, catecholamines, and oxygen consumption (myocardial perfusion, partial oxygen pressure, glucose substrate). Since subendocardial fibers (therefore longitudinal) are more sensitive to hypoperfusion [45], this mechanism cannot be incriminated in GD cardiac dysfunction, but diabetes-induced hypoxemia [13] and increased oxidative stress [11] could explain the reduced contractility. However, the predominant effect of GD on the transverse contractility is yet to be explained, and the particularities of RV biomechanics could bring more insight into this matter.

The fact that our study shows significant contractility changes only for the mid and apical ventricular regions is consistent with the fact that myocardial deformation is generally more pronounced in the apical segments than the basal region [26] and could be partly explained by the fibrous structure of the atrioventricular roof.

### 4.2. Technical Issues and Measurement Reliability

Our study found that FetalHQ^®^ can be used to assess simultaneously the biventricular shape, size and contractility, with moderate to good reliability, even at a frame rate of <80 Hz, but with poorer repeatability for the RV, in general, and for the 24-segment analysis, in particular.

Speckle-tracking technology is deemed as angle-independent, but it would be more accurate to say it is less angle-dependent than conventional functional echocardiography. For optimal endocardial border identification, the IVS should be oriented in a horizontal or oblique direction. Even so, the papillary muscles and the moderator band of the RV significantly hamper the endocardial tracking, and manual adjustments must almost always be performed, sometimes even multiple times if the superimposed tracking curves do not match the endocardium in the cine-loop.

The recommended 80 Hz frame rate threshold [46] significantly limits the use of FetalHQ^®^. Despite our best efforts to obtain an optimal image, this threshold was achieved in only 12% of the GD scans and 9% of the control scans, with a mean frame rate of 65–68 Hz. Although this does not meet the proprietary recommendations, it reflects the practical aspects of translating this technology into clinical setting. By using the ultrasound machines of a regular hospital, only one in ten scans will achieve a frame rate higher than 80 Hz in a population with a normal BMI (<25 kg/m^2^). Thus, most patients would not be eligible for FetalHQ^®^ analysis. This brings into question the applicability of FetalHQ^®^ in evaluating GD outside of research purposes. Moreover, from our experience, a high frame rate does not necessarily guarantee good endocardial visualization, and it is equally important to have an adequate amniotic fluid pocket between the uterine wall and the fetal thorax. Therefore, one can achieve a better image in an obese patient with hydramnios than in a thin patient with a fetus curled up against an anterior placenta.

As for the technique reliability, studies using frame rates higher than 80 Hz reported excellent intra- and inter-observer repeatability for GLS [7,21] and FAC [21] for both ventricles. In this study, the intra-rater and inter-rater reliability for LV global measurements ranged from good to excellent, but RV GLS and FAC had poor reliability. Nogué et al. reported poor reliability for the 24-segment SI and FS at a 60 Hz threshold, especially for the basal segments [47]. In our study, reliability was moderate to good for EDD and SI for both ventricles, but the FS had poor repeatability for the RV, especially for segment 1. Moreover, in comparison to the LV, the intra-rater and inter-rater ICC were lower for most RV measurements, presumably because of the moderator band and of the three papillary muscles.

### 4.3. Study Strengths

The main strengths of this study are its rigorous methodology, and a homogenous population including only GD fetuses diagnosed according to the novel criteria [1] that were compared to healthy fetuses examined at a GA-difference of 2 days, maximum. Thus, we reduced the selection bias by recruiting healthy fetuses mainly based on their GA. We performed a comprehensive simultaneous assessment of the geometry and contractility of both ventricles, using the 24-segment analysis. By removing the effect of in-plane motion (the tangential movement of the fetal heart), two-dimensional speckle tracking allowed us to measure the same segments in systole and diastole [28].

### 4.4. Study Limitations

The small number of individuals included in our study is a significant limitation, as well as the recruitment of patients from the private sector. Such patients usually benefit from better access to healthcare and have a close, direct relationship with their primary obstetrician, which tends to promote a better follow-up, timely delivery, and lower adverse outcome risk, even though a significant proportion of our patients did not strictly control their diabetes. Thus, further studies on a larger sample size recruited from the public healthcare system are needed to confirm our study’s conclusions.

The use of specialized, expensive software is another limitation for transposing this technique into clinical practice. However, the most important limitation of our study is lowering the frame rate threshold below 80 Hz, which could be responsible for the lower ICC values. This could also be viewed as a strength, since significant cardiac changes have been detected, even for lower quality 4C cine-loops, and this image quality better reflects the reality of day-to-day obstetrical care. Another shortcoming is the influence of through-plane motion on two-dimensional speckle tracking. However, we used 4C views, which are less affected by this displacement than short-axis views [48]. Furthermore, some FetalHQ^®^ measurements were inconsistent with previous speckle-tracking studies. This could be explained by the different study design, GD definition, diabetes control, image quality and different speckle-tracking software.

### 4.5. Clinical Impact

This study contributes to the accumulating evidence of subclinical cardiac dysfunction in GD-exposed fetuses by demonstrating decreased contractility of the RV mid and apical segments. The fact that there is a Z-score difference of more than 1 SD supports the practical relevance of these changes. The clinical impact of this finding could be double fold: improving perinatal outcome by identifying fetuses at high risk of stillbirth, and thus, scheduling their delivery in due time at an appropriate facility, and improving long-term outcome by identifying individuals at risk for cardiovascular disease. Studies suggest that diabetes-related cardiac function changes persist in the neonatal period [19,33], but generally normalize over time [49,50]. However, Schütte et al. demonstrated that male adult offspring from type 2 diabetic rats develop cardiac dysfunction only after a hyper-fat diet [51]. It is generally thought that fetal programming is mediated by epigenetic changes in the expression of specific genes, and various patterns of DNA methylation have indeed been incriminated in ‘malprogramming’ diabetes-exposed fetuses [52]. Thus, one could argue the fetus is programmed to manifest cardiac impairment later in life only under certain conditions, therefore in utero exposure to diabetes should prompt a more controlled diet and appropriate follow-up for cardiovascular risk evaluation.

## 5. Conclusions

FetalHQ^®^ can be used for a fast and simultaneous assessment of the shape, size, and contractility of both ventricles, with moderate to good reliability, even at lower frame rates, but with poorer repeatability for the RV, in general, and for the 24-segment analysis, in particular. RV transverse contractility is impaired in GD-exposed fetuses, especially in the mid and apical segments.

Longitudinal studies using novel echocardiography techniques could explain the mechanism of fetal cardiac dysfunction, and subgrouping by diabetes onset and glycemic control, from fetus to infant to adulthood, could bring more insight into fetal programming and adult disease.

## Data Availability

The data used to support the findings of this study are available upon request to the corresponding author.

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
