# Peer review of "Evaluation of Fetal Cardiac Geometry and Contractility in Gestational Diabetes Mellitus by Two-Dimensional Speckle-Tracking Technology"

_diagnostics, 2022, doi:10.3390/diagnostics12092053_

Round 1
Reviewer 1 Report
This study aimed to assess the effect of gestational diabetes mellitus (GD) on fetal cardiac geometry and contractility by speckle tracking technology. In particular, the authors compared the fetuses data obtained from 33 pregnant patients with GD with the data obtained from 30 healthy pregnant. Based on the results, fetuses exposed to GD present impaired right ventricular contractility, especially in the mid and apical segments.
The study is well described. Moreover, the authors included the strength and limitations of the study.
I have only a few suggestions before publication. I suggest improving the discussion section by comparing their results with international data. Moreover, I suggest transferring the paragraph "Future research" to "Conclusion".
Author Response
Dear Reviewer,
The authors would like to thank you for accepting to review this manuscript and for your thoughtful suggestions.
We have improved the 'Discussion' section by comparing our results with previous data, as well as by presenting more data on potential pathophysiological mechanisms. We have also changed the 'Future studies' section with 'Clinical impact', and future research suggestions were moved to 'Conclusion'.
We are ready to do more work on the manuscript if the revisions are not yet entirely satisfactory.
Sincerely,
Roxana Gireadă
Reviewer 2 Report
In this study, the authors evaluated the effect of gestational diabetes on cardiac geometry and contractility. The manuscript has a number of strengths and several shortcomings that need to be corrected before publication.
Strengths of the manuscript:
- practical significance,
- the manuscript is well structured,
- methodology and statistical analysis are qualitative,
- the authors outlined the advantages, limitations and future prospects.
Disadvantages:
- the chapter "introduction" is uninformative. The authors need to describe the pathophysiological mechanisms (how does gestational diabetes mellitus cause subclinical cardiac dysfunction and hypertrophic cardiomyopathy?)
- a small number of patients.
- why were the levels of glycosylated hemoglobin (HbA1c) not determined?
- insufficient number of figures. I recommend adding figures with pathophysiological molecular mechanisms and contractile disorders.
Author Response
Dear Reviewer,
The authors would like to thank you for accepting to review our manuscript and for you thoughtful comments.
1. We have improved the 'Introduction' section by adding potential pathophysiological mechanisms of fetal subclinical cardiac dysfunction and hypertrophic cardiomyopathy in gestational diabetes mellitus.
2. The small number of patients has several explanations:
- we used FetalHQ software, that depends on an expensive Voluson E10 machine, and in our city it is available only in a handful of private practice offices; therefore, machine and software access were quite limited;
- recruiting was done from the private sector, where patients are more reluctant to participate to prospective studies;
- FetalHQ analysis can be tedious, and previous studies have evaluated 49 [Huang 2022] or 58 GD fetuses [Wang 2021]. For statistical reasons, our aim was to include at least 30 gestational diabetes cases.
3. Why were the levels of glycosylated hemoglobin (HbA1c) not determined? We preferred to use capillary glucose values in the last 10 days prior to the scan in order to characterize diabetes control in our population (and have added this aspect to Methodology, rows 104-106), since this is preferred by Romanian diabetologists, and we also believe it reflects better the momentary fetal cardiac function (since our primary outcome was not IVS hypertrophy, which is a structural parameter).
4. Insufficient number of figures. We have added a figure that compares right ventricular fractional shortening between gestational diabetes and healthy fetuses. We believe figures with pathophysiological molecular mechanisms and contractile disorders could not be sustained only by our findings, but would be more appropriate in a review of fetal cardiac function in gestational diabetes.
We are ready to do more work on our manuscript if the revisions are not entirely satisfactory, especially in what relates to figures.
Sincerely,
Roxana Gireada
Reviewer 3 Report
1. The English need improvement since there are some grammatical and syntax errors in the manuscript. For example,
· in line number 36, the words “a hyperglycemia” may be as “hyperglycemia”;
· in line number 43, “and to” as “and”;
· in line number 136, “amniotic” as “the amniotic”;
· in line number 142, “Gain” as “The gain”;
· in line number 201, “chi-square” as “the chi-square”;
· in line number 227, “11/13 patients” as “of 11/13 patients”;
· in line number 242, “differences of” as “differences in”;
· in line number 274, 275 and 276 , “was” as “were”;
· in line number 343, “that were” as “were”.
The grammar mistakes which are not mentioned here are also to be checked and corrected properly.
2. There are some typing mistakes as well, and authors are advised to carefully proof-read the text. For example,
· all over the manuscript, the word “cineloop” may be as “cine-loop”;
· in line number 42, “cases the” as “cases, the”;
· in line number 94, “3rd trimester” as “3rd-trimester”;
· in line number 95, “Multiple pregnancy” as “Multiple pregnancies”;
· in line number 103, “were were” as “were”;
· in line number 151, “Global” as “The global”; all over the manuscript, “atrio-ventricular” as “atrioventricular”;
· in line number 188, “first trimester” as “first-trimester”;
· in line number 205, “p value” as “p-value”;
· in line number 254 and 294, “24-segments” as “24-segment”;
· in line number 256, “presents the presents the” as “presents the”;
· in line number 260, “the segments” as “segments”;
· in line number 290, “crescent-shape” as “crescent shape”.
The typos not mentioned here are also to be checked and corrected properly.
3. Check the abbreviations throughout the manuscript and introduce the abbreviation when the full word appears the first time in the text and then use only the abbreviation (For example, gestational diabetes mellitus (GD), correlation coefficient (ICC), etc.,). And it should be in both abstract as well as in the remaining part of the manuscript. Make a word abbreviated in the article that is repeated at least three times in the text, not all words need to be abbreviated.
4. In the introduction, the authors should give the recent data about the prevalence of gestational diabetes mellitus, since the reference cited is 2014. And also the year it wrongly cited, instead of “2014” the authors have mentioned as “2004”.
5. In materials and methods, the authors should clearly mentioned about the control cases (healthy patients), since either based on only glucose level or any other criteria are followed for better understanding.
6. The authors are encouraged to modify the word “healthy patients” as “healthy individuals” all over the manuscript.
7. The authors may improve the discussion of their work by focusing on the present findings and introducing data from other authors who also worked with the same or other studies with recent references.
Author Response
Dear Reviewer,
The authors would like to thank you for accepting to revise our manuscript and for you thoughtful suggestions.
1-3. English grammar/syntax errors, typing mistakes and abbreviations have been revised and corrected as you suggested.
4. Recent data about the prevalence of gestational diabetes mellitus - we have reported the weighted prevalence of GD in Europe extracted from the supplementary material of a 2021 meta-analysis (Rows 38-39) [Paulo, M.S.; Abdo, N.M.; Bettencourt-Silva, R.; Al-Rifai, R.H. Gestational Diabetes Mellitus in Europe: A Systematic Review and Meta-Analysis of Prevalence Studies. Front Endocrinol 2021, 12, 691033. https://doi.org/10.3389/fendo.2021.691033].
5. The control group was clearly defined as individuals having a normal 2-hour 75g oral glucose tolerance test at 24-28 weeks of gestation and no suspicion of gestational diabetes throughout the remainder of the pregnancy (Rows 101-103).
6. “Healthy individuals” syntagm was used all over the manuscript.
7. The authors may improve the discussion of their work by focusing on the present findings and introducing data from other authors who also worked with the same or other studies with recent references. We have significantly improved the Discussions, by introducing more data from previous studies, and by presenting fetal right ventricle particularities and potential pathophysiological mechanisms for systolic dysfunction (also referenced in the Introduction).
We are ready to do more work on our manuscript it the revisions are not yet entirely satisfactory.
Sincerely,
Roxana Gireadă
Round 2
Reviewer 2 Report
Dear Authors, Thank You so much for Your reply and corrections.
The authors have improved the manuscript, however, I still have some questions:
1) the authors should add a chapter "limitations" and note a number of points (for example, a small number of patients). Thus, further studies with a large number of patients are needed to confirm the author's conclusions.
2) I believe and insist that a figure/diagram with pathophysiological mechanisms should be added.
Best Regards, 17/08/2022
Author Response
Dear Reviewer,
Thank you for answering us in such short notice!
- Limitations - we have added them as a separate chapter, including the small number of patients in our shortcomings (Rows 445-462).
- Pathophysiology figure - we have embedded an original figure in the Introduction section (Row 68).
We believe these last changes have really improved our manuscript. If readjustments are necessary, we are happy to oblige.
Sincerely,
Roxana Gireada